# Defective BVES-mediated feedback control of cAMP in muscular dystrophy

Haiwen Li [1,2], Peipei Wang [1], Chen Zhang[1,2], Yuanbojiao Zuo[1,2], Yuan Zhou [1,2] & Renzhi Han [1,2] ✉

Biological processes incorporate feedback mechanisms to enable positive and/or negative regulation. cAMP is an important second messenger involved in many aspects of muscle biology. However, the feedback mechanisms for the cAMP signaling control in skeletal muscle are largely unknown. Here we show that blood vessel epicardial substance (BVES) is a negative regulator of adenylyl cyclase 9 (ADCY9)-mediated cAMP signaling involved in maintaining muscle mass and function. BVES deletion in mice reduces muscle mass and impairs muscle performance, whereas virally delivered BVES expressed in *Bves*-deficient skeletal muscle reverses these defects. BVES interacts with and negatively regulates ADCY9's activity. Disruption of BVES-mediated control of cAMP signaling leads to an increased protein kinase A (PKA) signaling cascade, thereby promoting FoxO-mediated ubiquitin proteasome degradation and autophagy initiation. Our study reveals that BVES functions as a negative feedback regulator of ADCY9-cAMP signaling in skeletal muscle, playing an important role in maintaining muscle homeostasis.

cAMP is an important second messenger mediating the signaling cascade of numerous G protein-coupled receptors (GPCRs) involved in various aspects of muscle physiology, such as glycogenolysis, contractility, sarcoplasmic calcium dynamics, and muscle mass maintenance[1]. It is synthesized by ADCYs and degraded by phosphodiesterases (PDEs)[1]. As many biological processes have evolved feedback mechanisms to enable positive and/or negative regulation[2], the feedback mechanisms for the cAMP signaling control in skeletal muscle, however, are largely unknown.

Previous studies reported that several effector proteins can bind cAMP with high affinity, such as PKA, exchange factor directly activated by cAMP (EPAC), hyperpolarization-activated cyclic nucleotide-gated channels (HCN) and the Popeye domain containing (POPDC) family[3,4]. As a class of membrane-localized cAMP-binding proteins, POPDC proteins including POPDC1 (commonly known as BVES), POPDC2 and POPDC3, are abundantly expressed in skeletal muscle and heart[5–7]. Genetic mutations in *BVES* were identified in patients with limb-girdle muscular dystrophy type R25 (LGMDR25) and cardiac

arrhythmia[3,8–14,12,15]. Similar muscle and heart dysfunction was reported in mice[16,17], zebrafish[8], Xenopus[18] and Drosophila[19] models with *BVES* deficiency, suggesting that BVES plays important, highly conserved functions in striated muscles.

In this study, we employed genetic, pharmacological, biochemical and live cell imaging approaches to study the molecular pathogenesis of BVES-deficient muscular dystrophy. Our studies unveiled a role of BVES in providing a negative feedback control for ADCY9 to regulate the cAMP signaling in skeletal muscle. The loss of BVES-mediated feedback control of the cAMP signaling promoted muscle atrophy and dysfunction.

## Results

### *Bves* ablation impairs muscle function and exercise performance in mice

To investigate the physiological role of BVES in skeletal muscle, we analyzed the expression of *Bves* in various mouse tissues by RT-PCR and found that the *Bves* transcript was highly expressed in the striated

---

[1]Department of Pediatrics, Herman B Wells Center for Pediatric Research, Indiana University School of Medicine, Indianapolis, IN 46202, USA. [2]Department of Surgery, Davis Heart and Lung Research Institute, Biomedical Sciences Graduate Program, Biophysics Graduate Program, The Ohio State University Wexner Medical Center, Columbus, OH 43210, USA. ✉e-mail: rh11@iu.edu

muscle (Supplementary Fig. 1a), consistent with previous reports[20]. Next, we established a *Bves* knockout (BVES-KO) mouse line, in which the entire coding region of *Bves* spanning exon 2 to exon 8 were deleted (Supplementary Fig. 1b). Immunostaining showed that BVES was mainly localized at the sarcolemma of wild-type (WT) muscle fibers and its expression was completely disrupted in the skeletal muscle of BVES-KO mice (Fig. 1a). Consistently, muscles of BVES-KO mouse showed remarkable decrease at both *Bves* transcript (Supplementary Fig. 1c) and protein levels including monomeric, glycosylated and dimeric forms (Supplementary Fig. 1d, e). Interestingly, the transcript and protein expression of *Bves* were decreased by about 50% in the muscles of *Bves* heterozygous mice compared with WT (Supplementary Fig. 1d, e). However, the transcript expression of the other two members of the POPDC family including *Popdc2* and *Popdc3* was not significantly changed in skeletal muscles of BVES-KO mice (Supplementary Fig. 1f).

The BVES-KO mice were fertile and smaller compared with the age/sex-matched littermate controls (Fig. 1b). The male BVES-KO mice showed a retarded growth from 3 months of age compared with the age/sex-matched WT mice (Fig. 1c). The female BVES-KO mice showed similar retarded growth but with a delayed onset (Supplementary Fig. 1g). Kaplan–Meier survival curve revealed that the male BVES-KO mice had a reduced life span with 50% survival by around 60 weeks of age (Fig. 1d). To evaluate if *Bves* disruption affects the physical performance, we subjected the mice to voluntary wheel running for 9 consecutive days. As shown in Fig.1e, the BVES-KO mice showed reduced running distance compared to WT littermate controls. Similarly, BVES-KO mice displayed a remarkable decrease in total running distance and the time to exhaustion in forced treadmill running test (Fig. 1f, g). The BVES-KO mice displayed more dropouts, particularly at higher running speeds, than WT mice (Fig. 1h). To test if *Bves* disruption compromised the muscle function, we measured the muscle contractility using an in vivo muscle test system[21]. Maximum plantarflexion tetanic torque was measured during supramaximal electric stimulation of the tibial nerve at 150 Hz. The BVES-KO mice exhibited progressive loss of muscle contractile strength starting from around 4 months of age (Fig. 1i). Interestingly, the heterozygous BVES-KO mice also displayed a significant loss of force production at 12 months of age but not at 5 months of age (Supplementary Fig. 1h), indicating that haploinsufficiency of BVES compromises muscle function. Taken together, these results suggest that BVES plays an important role in maintaining muscle function.

## *Bves* ablation leads to muscular dystrophy and atrophy

Next, we performed histopathological analysis of skeletal muscle in BVES-KO and WT mice. Reduced muscle mass was clearly visible in 4-month-old BVES-KO mice as compared to the age-matched WT littermate controls (Fig. 2a). At both 4 and 8 months of age, the mass of various skeletal muscles including gastrocnemius (GA), quadriceps (QU) and tibial anterior (TA) was significantly decreased (Fig. 2b, c, Supplementary Fig. 2a), while the soleus muscle showed a trend of reduction (Supplementary Fig. 2b). The heart mass was decreased in BVES-KO mice at 8 months of age (Supplementary Fig. 2c). Since the BVES-KO mice were overall smaller than WT, we also calculated the muscle mass normalized to body weight. Again, we found that the normalized muscle mass of GA and QU in BVES-KO mice was still significantly decreased as compared with WT littermates (Fig. 2d, e) while the normalized mass of TA, soleus and heart showed no significant changes or slight increase in BVES-KO mice (Supplementary Fig. 2d–f). Similar results were observed in female mice (Supplementary Fig. 3), suggesting that the loss of BVES causes muscle atrophy regardless of gender.

Muscle necrosis, centrally nucleated muscle fibers (CNFs) and angulated muscle fibers were readily observed in the H&E-stained

sections of GA muscles from BVES-KO mice at 4 and 8 months of age (Fig. 2f, Supplementary Fig. 4). The fiber size distribution of the GA muscle shifted to the smaller fiber side in both 4- and 8-month-old BVES-KO mice compared with WT (Fig. 2g, Supplementary Fig. 4). The mean cross-sectional area (CSA) of muscle fibers were significantly reduced in both 4- and 8-month-old BVES-KO mice compared with WT (Fig. 2h). The percentage of CNFs was increased to $12.8 \pm 0.8$ % and $17.0 \pm 3.0$ % in 4- and 8-month-old BVES-KO muscles, respectively (Fig. 2i). Serological analysis showed that the serum level of muscle creatine kinase was significantly elevated in BVES-KO mice compared with WT (Fig. 2j), consistent with previous reports in human patients with *BVES* mutations[22,23]. These results suggest that muscular atrophy occurs along with dystrophy in BVES-KO muscles.

We further studied the impact of BVES deficiency on different types of fibers in GA muscle. We performed immunofluorescence staining with antibodies against different isoforms of myosin heavy chain and dystrophin. As shown in Fig. 2k, l, the type IIb muscle fibers were significantly smaller in the BVES-KO mice than those in WT mice, while the type I and IIa muscle fibers were less affected. There was no significant difference in the percentage of each muscle fiber types in BVES-KO GA muscles as compared to the WT controls (Fig. 2m). In contrast to mice, human skeletal muscle contains mainly IIx instead of IIb fibers, despite their similarity in biochemical properties[24]. Previous studies showed that mouse *flexor digitorum brevis* (FDB) muscle consisted of around 50% type IIx muscle fibers[25]. To examine if type IIx muscle fibers are also affected in BVES-KO mice, we performed the immunostaining with anti-MyHC-2a and anti-MyHC-1 as well as anti-dystrophin antibodies (Supplementary Fig. 5a). The MyHC-1 and 2a double negative fibers should be type IIx as type IIb fibers are rarely seen in FDB muscles (data not shown). Quantitative measurements showed that the mean CSA of type IIx fibers was also significantly decreased in BVES-KO FDB muscles as compared with WT controls (Supplementary Fig. 5a–c).

To test if the muscular dystrophy and atrophy in BVES-KO mice were due to the specific loss of BVES in skeletal muscle, we performed a rescue experiment with adeno-associated virus 9 (AAV9)-mediated gene transfer of BVES in skeletal muscle. We generated an AAV9 carrying human *BVES* cDNA fused with the HA tag under the control of MHCK7 (Fig. 3a), a muscle-specific promoter active in mature skeletal muscle but not in muscle satellite cells[26]. AAV9-BVES ($2 \times 10^{11}$ vg) was injected into the GA muscles of 3-month-old BVES-KO mice. At 1 month after AAV9-BVES injection, immunofluorescence staining showed that almost all muscle fibers were positive for BVES (Fig. 3b). Similar to endogenous BVES (Fig. 1a), the BVES transgene expression was also mainly localized at the sarcolemma (Fig. 3b). Western blot analysis confirmed that BVES-HA transgene was highly expressed in the AAV9-BVES treated GA muscles but not in the contralateral GA muscles (Fig. 3c). Delivery of AAV9-BVES significantly increased muscle force production in BVES-KO mice at 1 month after injection and the rescue effect of AAV9-BVES on muscle force became more evident at three months (Fig. 3d). Moreover, AAV9-BVES delivery significantly increased the GA muscle mass by ~15% at 1 month and ~47% at 3 months after injection compared with the contralateral untreated GA muscles (Fig. 3e), but the net muscle mass in AAV9-BVES treated GA muscle remained lower than that in WT mice (Supplementary Fig. 6). There was no significant change in the mass of the non-injected QU muscle (Fig. 3f). H&E staining showed that the muscle pathology was remarkably improved in the GA muscles treated with AAV9-BVES (Fig. 3g). The muscle fiber size distribution was normalized (Fig. 3h), and the average muscle fiber size was increased by ~19% following AAV9-BVES treatment (Fig. 3i). The percentage of CNFs was dramatically decreased in AAV9-BVES-treated GA muscles (Fig. 3j). These results suggest that the muscle pathology in the BVES-KO mice can be largely attributed to the loss of BVES expression in mature skeletal muscle fibers.

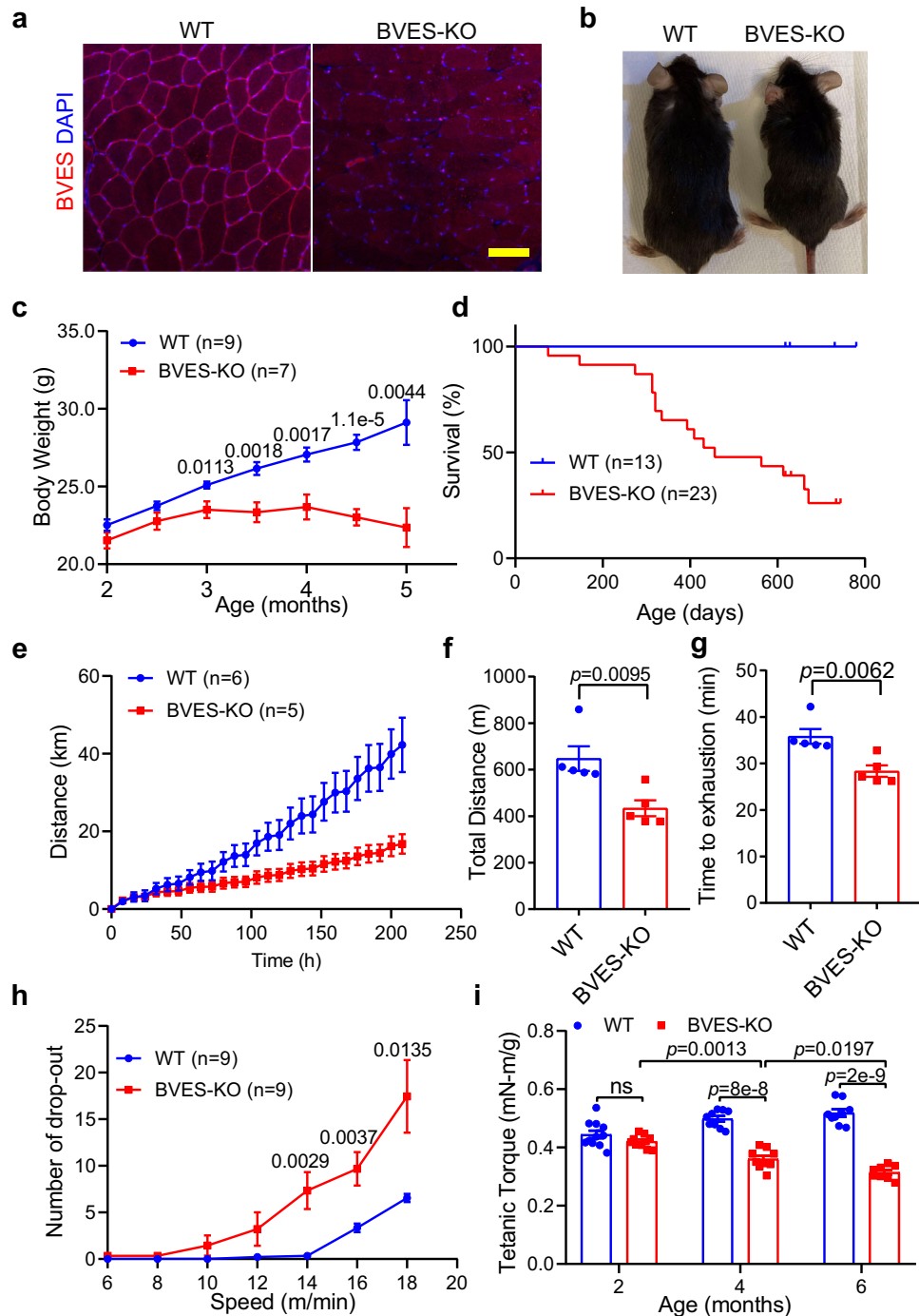

**Fig. 1 | BVES disruption compromises the body weight gain and muscle function in mice.** All animal experiments were performed in WT and BVES-KO male mice with the C57BL/6N genetic background. **a** Immunofluorescence images of GA muscles in WT and BVES-KO mice (4 months of age) stained with the antibody against BVES and DAPI. Scale bar: 100 μm. ($n = 4$ per genotype). **b** Representative image of WT and BVES-KO male littermates at 4 months of age. **c**, Body weight gain of male BVES-KO and age/sex-matched WT mice from two to five months of age. Two-tailed paired Student's *t* test. **d** Kaplan–Meier survival curve of WT and BVES-KO male mice. **e** Voluntary wheel running of BVES-KO and age-matched WT male mice (4 months of age). **f**, **g** Endurance capacity test performed by treadmill running showing running distance (**f**) and time to exhaustion (**g**) in BVES-KO ($n = 5$) and WT ($n = 5$) male mice (4 months of age). Two-tailed unpaired Student's *t* test. **h** The number of dropouts to test the capacity of recovery from muscle injury on the treadmill in BVES-KO and WT male mice (6 months of age). Two-tailed paired Student's *t* test. **i** Tetanic torque measurements of the posterior compartment muscles of BVES-KO and WT male mice in age-dependent manner (2-month age: WT ($n = 11$), BVES-KO ($n = 9$); 4-month age: WT ($n = 9$), BVES-KO ($n = 9$); 6-month age: WT ($n = 9$), BVES-KO ($n = 8$)). ns indicates no significant difference. Two-way ANONA with Tukey's multiple comparisons test. Data are mean ± SEM. Source data are provided as a Source Data file.

## BVES interacts with and inhibits ADCY9-mediated cAMP signaling

To understand the mechanism by which BVES deficiency leads to muscular dystrophy and atrophy, we performed co-immunoprecipitation (co-IP) from AAV9-BVES-HA transduced skeletal muscle using anti-HA antibody followed by mass spectrometry to identify the BVES-interacting proteins (Fig. 4a). After subtracting the background from a control sample, we identified ~186 putative BVES-interacting proteins

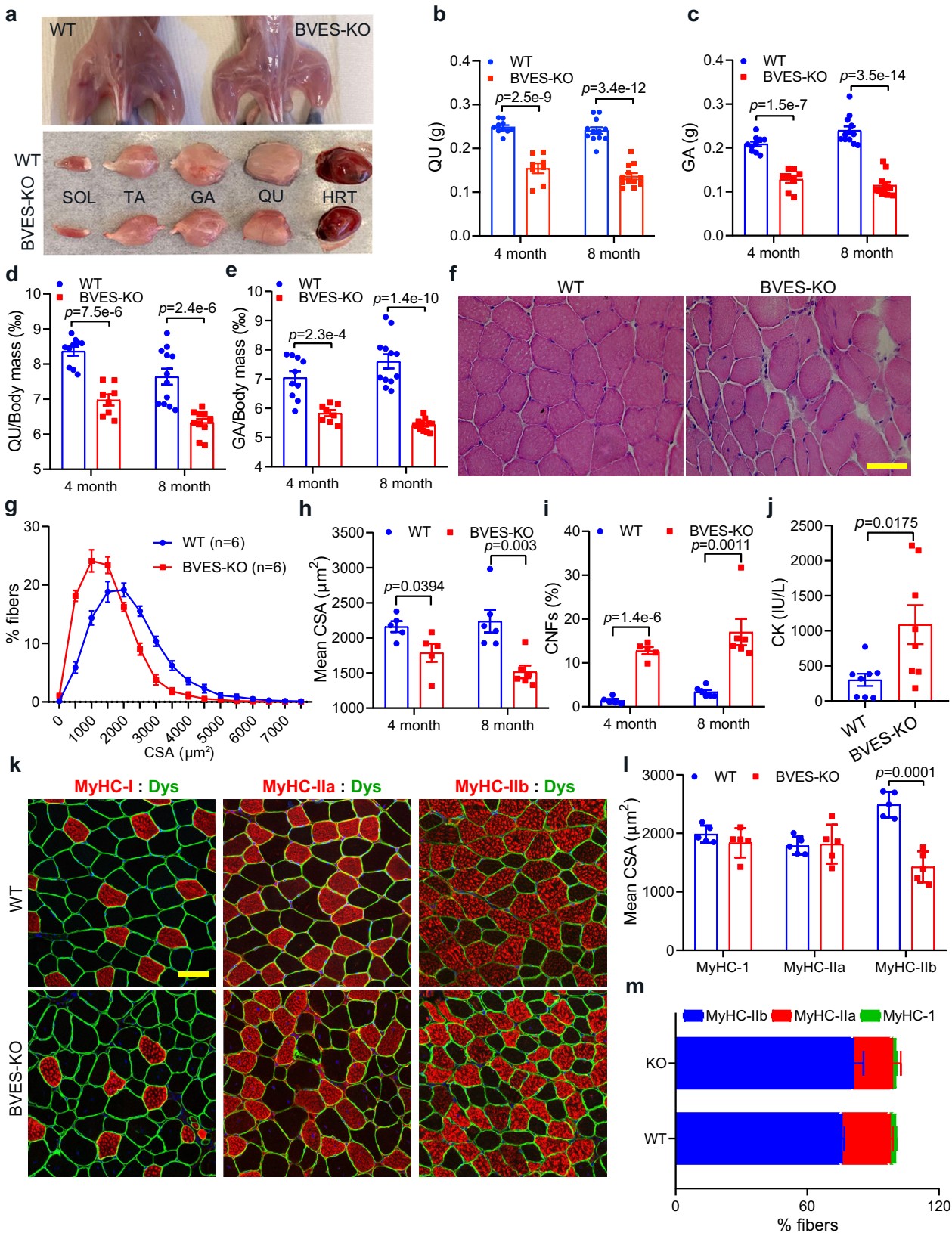

including some known interactors such as dysferlin and dystrophin[8] (Supplementary Data 1). Gene ontology (GO) analysis of subcellular localization found 73 candidates localized at the plasma membrane as the most highly enriched cellular compartment associated with BVES, 38 at endoplasmic reticulum (ER) and 16 at nuclear envelope, in concert with the fact that BVES was found at these subcellular compartments[4,27]

(Fig. 4b). GO analysis further showed that the BVES-interacting proteins are involved in several biological processes such as G protein-coupled receptor pathways, ion transport, membrane potential, muscle contraction, plasma membrane repair, muscle cell differentiation and nuclear envelope organization (Fig. 4c). ADCY9, a major adenylate cyclase responsible for cAMP biosynthesis in adult skeletal muscle[1], was

**Fig. 2 | BVES deficiency causes muscle atrophy and dystrophy in mice.** All animal experiments were performed in WT and BVES-KO male mice with the C57BL/6N genetic background. **a** Photographs of mice after the removal of skin and muscle tissues dissected from BVES-KO and WT male mice (4 months of age). **b, c** Net mass of QU (**b**) and GA (**c**) muscles in BVES-KO and WT male mice at 4 (QU or GA from WT: $n = 10$; KO: $n = 8$) and 8 (QU or GA from WT: $n = 12$; KO: $n = 12$) months of age. **d, e** The relative mass of QU (**d**) and GA (**e**) muscles normalized to the body weight at 4 (QU or GA from WT: $n = 10$; KO: $n = 8$) and 8 (QU from WT: $n = 12$; KO: $n = 11$; GA from WT: $n = 12$; KO: $n = 12$) months of age. **f** H&E-stained images of GA muscle cross-sections from 8-month-old BVES-KO and WT male mice. Scale bar: 100 μm. **g** The CSA distribution of GA muscle fibers in 8-month-old BVES-KO and WT male mice. CSA was quantified from the entire cross-sections of GA muscles. **h** Mean CSA of GA muscle fibers at 4 ($n = 5$ for WT and KO) and 8 ($n = 6$ for WT and KO) months

of age. **i** Quantification of CNFs in the male GA muscles at 4 ($n = 5$ for WT and KO) and 8 ($n = 6$ for WT and KO) months of age. **j** CK measurements in 8-month-old BVES-KO ($n = 8$) and WT ($n = 8$) male mice. **k** Representative immunofluorescence images of BVES-KO and WT male (8 months of age) GA muscle sections stained with dystrophin (Dys, green) and one of the myosin heavy chain antibodies (MyHC-I, MyHC-IIa and MyHC-IIb, red). Scale Bar: 100 μm. **l** Mean CSA of MyHC-I, MyHC-IIa and MyHC-IIb fibers in GA muscles from BVES-KO ($n = 5$) and WT ($n = 5$) male mice (8 months of age). CSA was quantified from the entire cross-sections of GA muscles. **m** The percentage of MyHC-I, MyHC-IIa and MyHC-IIb fibers in GA muscles from BVES-KO ($n = 5$) and WT ($n = 5$) male mice (8 months of age). Two-tailed unpaired Student's $t$ test. Data are mean ± SEM. Source data are provided as a Source Data file.

identified among the BVES-interacting proteins (Supplementary Data 1), which was further validated by co-IP with Flag-tagged ADCY9 and HA-tagged BVES in COS-1 cells (Fig. 4d).

We hypothesized that the cAMP-binding BVES may provide a feedback loop to regulate the cAMP biosynthesis via interacting with ADCY9. To test this hypothesis, we first generated a mutant HEK293 cell line deficient in ADCY3 and 6 (the two major ADCY genes expressed in HEK293 cells[28]) using cytosine base editing (CBE) (Supplementary Fig. 7), as a homologous reconstitution system to probe the function of exogenously expressed ADCY9. The ADCY3/6-mutant HEK293 cells were also stably transduced with cAMPr, a genetically engineered fluorescent sensor of cAMP[29]. Measurement of the cAMPr fluorescence in the ADCY3/6-mutant HEK293 cells showed that inhibition of PDEs with a non-selective PDE inhibitor 3-isobutyl-1-methylxanthine (IBMX) induced a gradual increase of cAMP in ADCY9-transfected cells but not in the mCherry-transfected cells (Fig. 4e), suggesting that the exogenously transfected ADCY9 contributed to the biosynthesis of cAMP in this mutant cell line. Interestingly, co-transfection with BVES and ADCY9 together resulted in a substantially reduced cAMP elevation in response to IBMX inhibition (Fig. 4e), suggesting that BVES inhibits ADCY9's activity. Consistently, the cAMP levels in the skeletal muscles were significantly increased in BVES-KO mice as compared to WT mice (Fig. 4f).

Elevation of cAMP activates PKA to mediate signal amplification[30]. Western blot showed the phosphorylation of PKA substrates was significantly increased in BVES-KO mice (Fig. 4g, h). Moreover, Western blot analysis of the downstream signaling of PKA revealed that phosphorylation of the PKA substrate LKB1 (liver kinase B1) and its primary target adenosine monophosphate (AMP)-activated protein kinase (AMPK) were significantly increased in BVES-KO GA muscles (Fig. 4i–k). Consistent with our phenotypical data (Fig. 2k–m), Western blot showed that AMPK phosphorylation was dramatically increased in *extensor digitorum longus* (EDL), a fast-twitch fiber predominant muscle, but not in the slow-twitch fiber predominant soleus muscle of BVES-KO mice (Supplementary Fig. 8a, b). Interestingly, the expression of BVES in EDL and soleus muscle of WT mice was not dramatically different (Supplementary Fig. 8c, d), suggesting that other regulatory mechanisms may exist in slow fibers to compensate for BVES deficiency. Together, our data suggest that the loss of BVES-mediated negative regulation of cAMP signaling leads to an increased PKA/AMPK signaling cascade, particularly in fast muscles.

## Dysregulated PKA signaling leads to activation of the ubiquitination-proteasome degradation system (UPS) in BVES-deficient skeletal muscle

Previous studies showed that AMPK activation promotes forkhead box (FoxO)-mediated UPS degradation[31–33], which may contribute to the muscle pathology in BVES-KO mice. Consistently, we found that the expression of FoxO1 and FoxO3a were significantly increased by over 2 folds in BVES-KO skeletal muscle compared with WT controls (Fig. 5a–c). Similarly, the FoxO-regulated ubiquitin E3 ligases Atrogin-1

and MuRF1 were also significantly upregulated at both the transcriptional (Supplementary Fig. 9) and protein levels in BVES-KO muscles (Fig. 5a, d, e). Moreover, the global protein ubiquitination was significantly increased in BVES-KO muscles both before (at 2 months of age) (Supplementary Fig. 10a, b) and after the disease onset (at 8 months of age) (Fig. 5f, g), suggesting that the UPS activation is likely a direct consequence of BVES deficiency but rather a secondary event following muscular dystrophy. Ubiquitin contains seven lysines (K6, K11, K27, K29, K33, K48 and K63) and ubiquitin molecules can be linked through any of these seven lysines. We examined the abundance of K48- and K63-linked ubiquitinated proteins in 8-month-old WT and BVES-KO skeletal muscles and found that there was no significant difference in either K48 or K63-linked ubiquitination in BVES-KO muscles compared with WT control (Supplementary Fig. 10c). This suggests that BVES deficiency may affect other lysines such as K11 and K29-linked ubiquitination, which are linked to proteosome degradation[34]. Consistent with the fact that the nuclear translocation is essential for FoxO to function as transcription factors, we observed that FoxO3 was significantly increased in the nuclei of BVES-KO skeletal muscles compared with WT (Fig. 5h, j). Conversely, AAV9-BVES gene delivery decreased FoxO3a in the nuclei of BVES-KO skeletal muscles (Fig. 5i, k). To further verify that PKA-mediated AMPK activation is involved in FoxO3 signaling, we performed chromatin immunoprecipitation (ChIP) assay with anti-FoxO3 antibody from BVES-KO skeletal muscle treated with or without H89, a specific inhibitor of PKA. PKA inhibition significantly suppressed FoxO3a binding to the promoters of MuRF1 and Atrogin-1 (Supplementary Fig. 11a–c). Collectively, these data suggest that cAMP/PKA/AMPK-mediated FoxO activation is involved in the activation of UPS in BVES-deficient skeletal muscle.

To further investigate the role of UPS in the pathogenesis of BVES-deficient muscular dystrophy and atrophy, we evaluated the effect of bortezomib, a selective inhibitor of 26 S proteasome approved by the U.S. Food and Drug Administration (FDA) to treat certain types of cancer such as multiple myeloma and mantle cell lymphoma[35–37], on the muscle function and pathology in BVES-KO mice. Bortezomib treatment significantly increased the body weight (Fig. 5l), enhanced the muscle contractility (Fig. 5m) and improved physical performance on treadmill running in BVES-KO mice (Fig. 5n, o). Moreover, bortezomib treatment significantly increased the mass of quadriceps and gastrocnemius muscles in BVES-KO mice (Fig. 5p, Supplementary Fig. 12). H&E staining showed that bortezomib treatment remarkably improved the muscle pathology as evidenced by more evenly organized muscle fibers (Fig. 5q). Bortezomib also significantly increased the fiber size (Fig. 5r, s) and reduced the percentage of CNFs (Fig. 5t). Taken together, bortezomib significantly improved muscle function and ameliorated the histopathology of skeletal muscles in BVES-KO mice.

As muscle mass is determined by the balance between protein synthesis and degradation[38], we also examined if BVES deficiency impacts protein synthesis using the SUnSET assay[39]. We found no significant changes in the total protein synthesis in BVES-KO skeletal

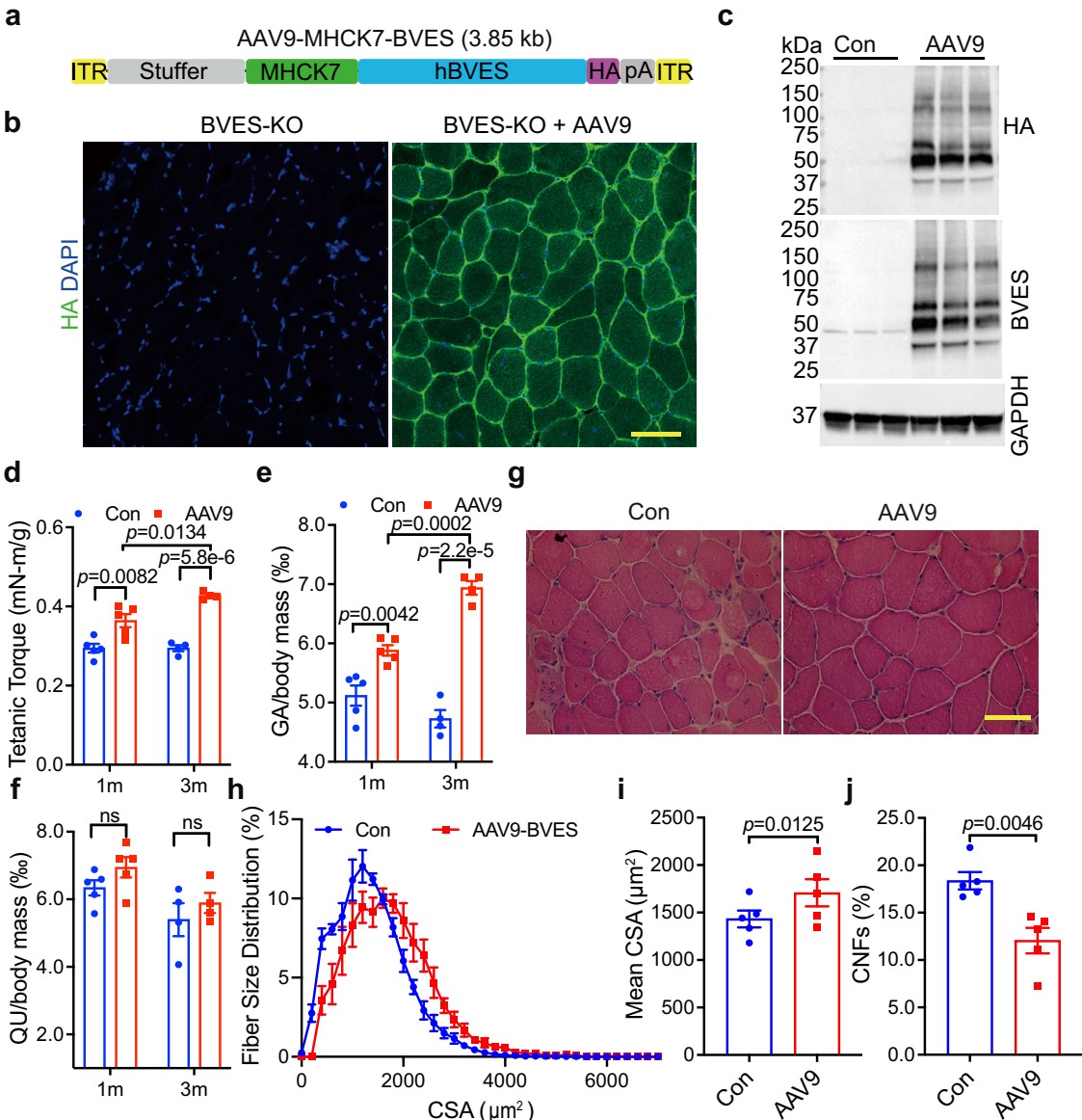

**Fig. 3 | Muscle-specific expression of BVES ameliorates skeletal muscle pathology in BVES-KO mice.** All animal experiments were performed in BVES-KO male mice with the C57BL/6N genetic background. **a** Schematic map of AAV9-BVES constructs. **b** Representative immunofluorescence images of AAV9-BVES treated and contralateral control GA muscles in BVES-KO male mice (5 months of age) stained with the antibody against HA-tag and DAPI. Scale bar: 100 μm. ($n = 6$ per treatment). **c** Western blot of BVES expression detected with anti-BVES and anti-HA antibodies in AAV9-BVES treated and contralateral control GA muscles (5 months of age). **d** Tetanic torque measurements of the posterior compartment muscles of AAV9-BVES treated and contralateral control GA muscles at 1 ($n = 5$ for Con and AAV9-BVES) and 3 ($n = 4$ for Con and AAV9-BVES) months after injection. **e, f** The relative mass of AAV9-BVES treated and contralateral control GA muscles (**e**) or untreated QU muscle (**f**) normalized by the body weight at 1 ($n = 5$ for Con and

AAV9-BVES) and 3 ($n = 4$ for Con and AAV9-BVES) months after injection. ns, not significant. Two-way ANONA with Tukey's multiple comparisons test (**d**–**f**). **g** H&E staining of AAV9-BVES treated or contralateral control GA muscle sections at 3 months after injection. Scale bar: 100 μm. **h** The CSA distribution of AAV9-BVES treated and contralateral control GA muscles from BVES-KO male mice at 3 months after injection. **i** Mean CSA of AAV9-BVES treated ($n = 5$) and contralateral control ($n = 5$) GA muscles from BVES-KO male mice at 3 months after injection. CSA was quantified from the entire cross-sections of GA muscles. **j** Quantification of CNFs in the AAV9-BVES treated ($n = 5$) or contralateral control ($n = 5$) GA muscles from BVES-KO male mice at 3 months after injection. CNFs was quantified from the entire cross-sections of GA muscles at 3 months after injection. Two-tailed unpaired Student's $t$ test (**i, j**). Data are mean ± SEM. Source data are provided as a Source Data file.

muscles (Supplementary Fig. 13a, b). We also examined the AKT/mTOR signaling pathway related to protein translation[40]. It was well known that phosphorylation of Thr308 (controlled by PDK1) and Ser473 (controlled by mTORC2) residues is required for maximal activation of AKT[41]. Interestingly, we observed that Thr308 phosphorylated AKT was significantly decreased while S473 phosphorylation was increased in BVES-KO muscles (Supplementary Fig. 13c–e). Furthermore, phosphorylation of both 4E-BP1 (Thr 37/46) and S6K (Thr389), the two downstream targets of mTOR that are responsible for protein

translation, was similar between WT and BVES-KO skeletal muscles (Supplementary Fig. 13c, f, g). These data suggest that the protein biosynthesis governed by the AKT/mTOR axis was minimally affected in BVES-KO mice.

**BVES deficiency compromises autophagy execution**
Since FoxO signaling also regulates autophagy-related genes, we reasoned that BVES disruption may alter the autophagy process. As expected, Western blot detected a significant upregulation of the

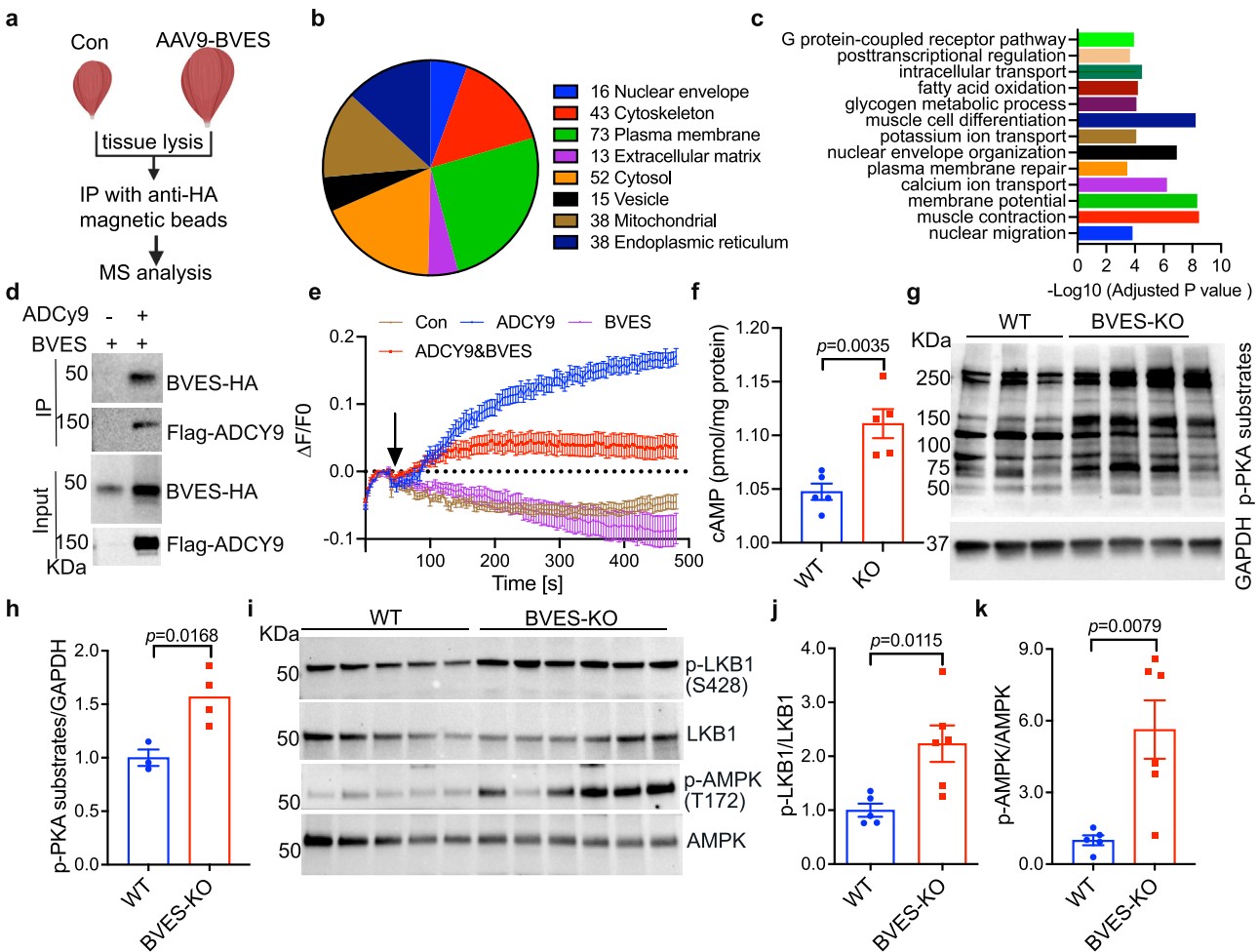

**Fig. 4 | BVES interacts with ADCY9 and regulates cAMP/PKA signaling.** All animal experiments were performed in WT and BVES-KO male mice with the C57BL/6N genetic background. **a** Diagram showing the IP-mass spectrometry approach to identify BVES-interacting proteins in mouse skeletal muscle (created with BioRender.com). **b, c** Gene ontology analysis of the cellular compartment (**b**) and biological process (**c**) categorization of co-immunoprecipitated proteins with BVES-HA. **d** ADCY9 was co-immunoprecipitated with BVES-HA using the Flag antibody from the lysates of Cos-1 cells co-transfected with Flag-ADCY9 and BVES-HA (experiments were repeated twice). **e** IBMX (500 μM)-induced cAMPr fluorescence changes in ADCY3/6 double mutant HEK293 cells with stable expression of cAMPr, transfected with the indicated plasmids. Cell number: 31, 15, 50, 30 for mCherry, BVES, ADCY9 and ADCY9 + BVES, respectively, from three independent trials per condition. **f** cAMP measurements in skeletal muscle lysates from BVES-KO ($n = 5$) and WT ($n = 5$) male mice (8 months of age). Two-tailed unpaired Student's $t$ test. **g, h** Western blot (**g**) and quantification (**h**) of p-PKA substrates in skeletal muscle from BVES-KO ($n = 4$) and WT ($n = 3$) male mice (8 months of age). Two-tailed unpaired Student's $t$ test. **i–k** Western blot (**i**) and quantification of phosphorylated LKB1(**j**) and phosphorylated AMPK (**k**) in skeletal muscle from BVES-KO ($n = 6$) and WT ($n = 5$) male mice (8 months of age). Two-tailed unpaired Student's $t$ test. Data are mean ± SEM. Source data are provided as a Source Data file.

autophagy markers LC3A, LC3B and p62 in the BVES-KO muscles (Fig. 6a–d). Likewise, immunostaining showed that p62 was accumulated in the BVES-KO skeletal muscles (Fig. 6e, Supplementary Fig. 14a). The transcript expression levels of the genes related to autophagy initiation such as *P62, Bnip3, Atg7, Cts1, Beclin-1* and *Park2* were also significantly increased in the BVES-KO muscles (Supplementary Fig. 14b), in line with the notion that their transcription is activated by the FoxO pathway[33]. Conversely, AAV9-mediated delivery of BVES reversed the aberrant changes in autophagy initiation associated with BVES deficiency (Supplementary Fig. 15). Furthermore, live cell imaging showed more RFP-LC3B+ puncta in the *flexor digitorum brevis* (FDB) muscle fibers of BVES-KO mice (Fig. 6h, i). Autophagy initiation is orchestrated by a number of protein complexes including the core negative regulator of autophagy (mTOR complex), ULK1 initiation complex (ULK1, FIP200, etc.), PI3K III nucleation complex (VPS15, VPS34, etc.), PI3P-binding complex (ATG16L, ATG5, etc.) and the protein related with lipid delivery (ATG9a)[42]. We found that VPS34 was dramatically increased in the BVES-KO muscles (Fig. 6f, g) while other proteins tested were unchanged (Supplementary Fig. 16).

To further understand how BVES ablation regulates the dynamic autophagy process, we next examined autophagic flux in WT and BVES-KO skeletal muscles using colchicine, which blocks the fusion of autophagosome with lysosome[43]. We observed a significant increase of LC3B upon colchicine treatment in WT muscles but not in BVES-KO muscles (Fig. 6j, k), indicating that BVES disruption suppresses the autophagic flux. This was corroborated by the live cell imaging study using the dual fluorescent reporter mCherry-GFP-LC3B[44] in FDB muscles of BVES-KO and WT mice. As shown in Fig. 6l, m, the percentage of the autolysosomes (indicated by the red-only puncta) was significantly reduced in BVES-KO mice.

The increased autophagy initiation and reduced autolysosome formation suggest that the fusion of autophagosome with lysosome is likely inhibited in the BVES-KO muscles. We performed Western blot to analyze the expression of some key proteins involved in the autophagic fusion such as the STX17, Rab7, VAMP7 and VAMP8. Interestingly, STX17 and VAMP7, but not the other proteins measured, were dramatically decreased by ~50% in BVES-KO muscles (Fig. 6n, o and Supplementary Fig. 16), suggesting that the suppression of autophagy

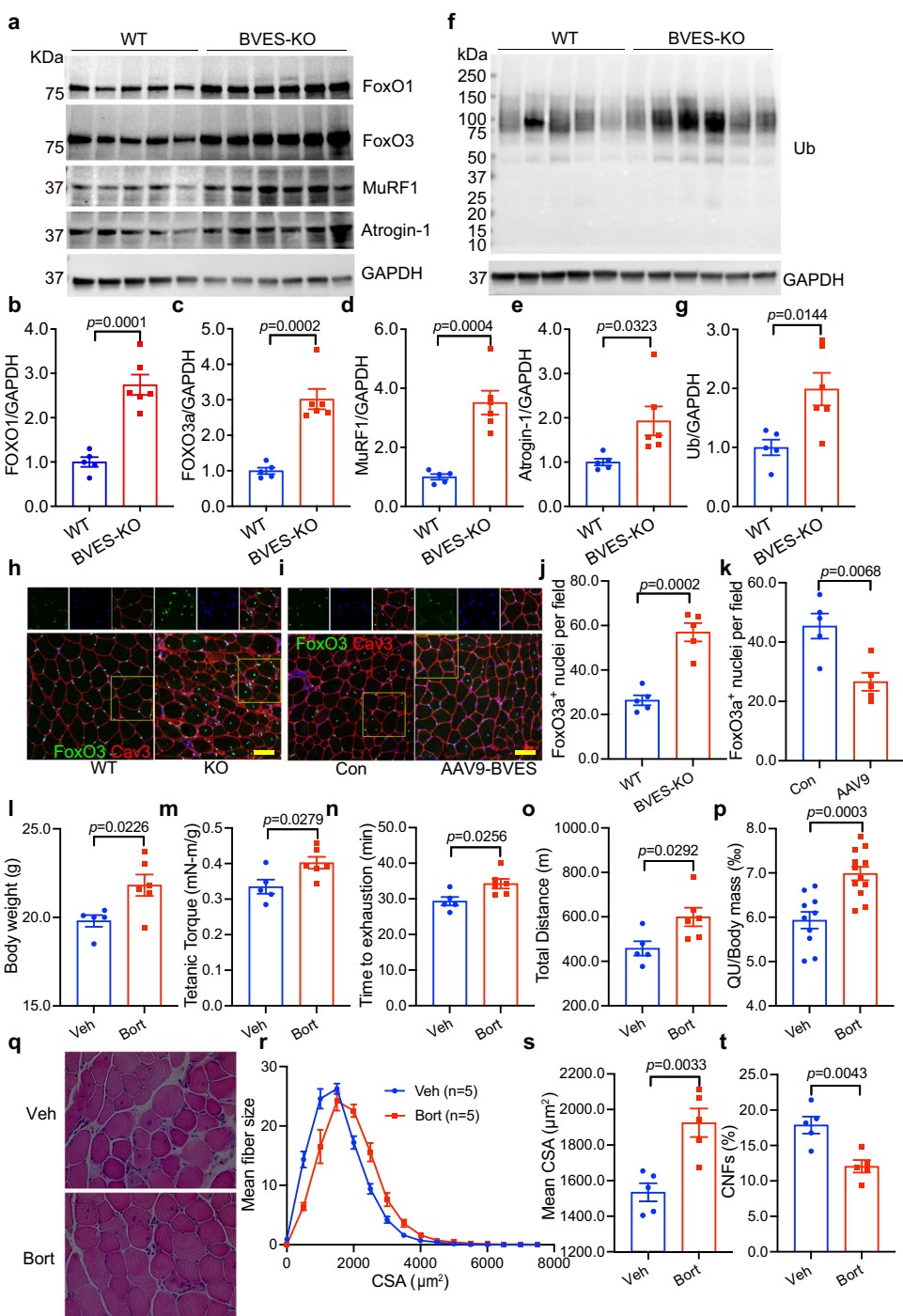

**Fig. 5 | BVES disruption enhances the UPS via FoxO.** All animal experiments were performed in WT and BVES-KO male mice with the C57BL/6N genetic background. **a**–**e** Western blot of FoxO signaling in total GA muscle extracts from 8-month-old WT ($n = 5$) and BVES-KO ($n = 6$) male mice (**a**) and quantification of FoxO1 (**b**), FoxO3a (**c**), MuRF1 (**d**) and Atrogin-1 (**e**) normalized to GAPDH. **f**, **g** Western blot (**f**) and quantification (**g**) of ubiquitination in total GA muscle extracts from 8-month-old WT ($n = 5$) and BVES-KO ($n = 6$) male mice. **h** Immunostaining of FoxO3a and Cav3 (labeling muscle fibers) of GA muscles from WT and BVES-KO male (4 months of age; $n = 5$ each). Scale bar: 50 μm. **i** Immunostaining of FoxO3a and Cav3 of GA muscles from AAV9-BVES treated and contralateral control GA muscles at 3 months after injection ($n = 5$ per treatment). Scale bar: 50 μm. **j, k** Quantification of FoxO3a positive nuclei per field in WT and BVES-KO ($n = 5$ each; 4 months of age) GA muscle (**j**), and AAV9-BVES treated and contralateral control GA muscles in BVES-KO male mice at 3 months after injection ($n = 5$ each) (**k**). Three random areas were quantified for each stained section per mouse. **l** Body weight of BVES-KO male mice at

2 months after vehicle ($n = 5$) or Bortezomib ($n = 6$) treatment. **m** Tetanic torque measurements of BVES-KO male mice at 2 months after vehicle ($n = 5$) or Bortezomib ($n = 6$) treatment. **n, o** Treadmill running test showing time to exhaustion (**n**) and total running distance (**o**) of BVES-KO male mice at 2 months after vehicle ($n = 5$) or Bortezomib ($n = 6$) treatment. **p** The normalized QU mass to body weight in BVES-KO male mice at 2 months after vehicle ($n = 10$) or bortezomib ($n = 12$) treatment. **q** Representative H&E staining of GA muscle sections from BVES-KO male mice at 2 months after vehicle or bortezomib treatment. **r** The CSA distribution of bortezomib treated and vehicle control GA muscles from BVES-KO male mice ($n = 5$ each) at 2 months after treatment. CSA was quantified from the entire cross-sections of GA muscles. **s** Mean CSA of bortezomib treated ($n = 5$) and vehicle control ($n = 5$) GA muscles from BVES-KO male mice at 2 months after treatment. **t** Quantification of CNFs in the bortezomib treated ($n = 5$) and vehicle control ($n = 5$) GA muscles at 2 months after treatment. Two-tailed unpaired Student's $t$ test. Data are mean ± SEM. Source data are provided as a Source Data file.

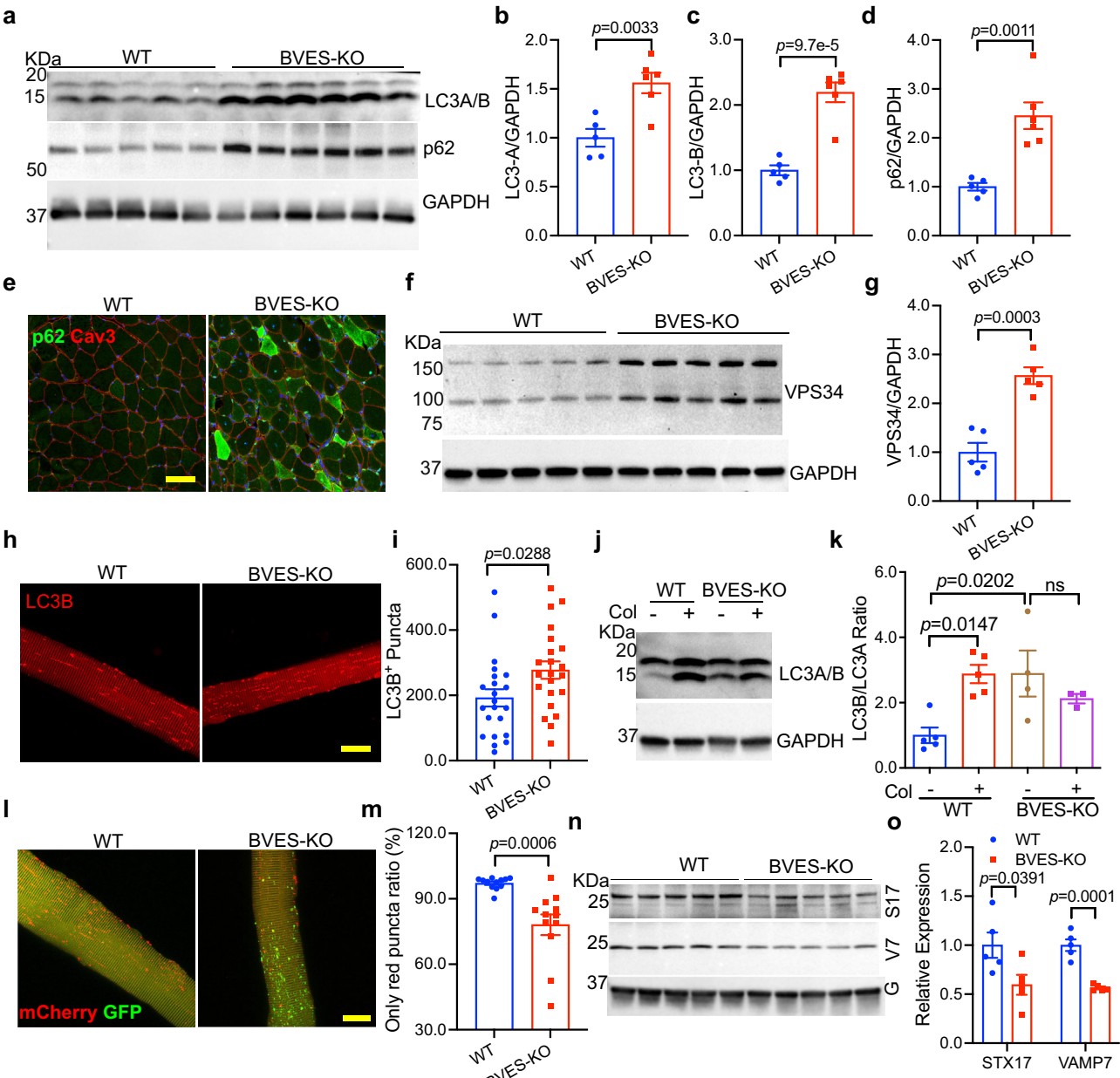

**Fig. 6 | BVES Disruption suppresses the autophagic execution.** All animal experiments were performed in WT and BVES-KO male mice with the C57BL/6N genetic background. **a–d** Western blot (**a**) of autophagic markers and quantification of LC3A (**b**), LC3B(**c**) and p62 (**d**) normalized to GAPDH in total GA muscle extracts from 8-month-old WT ($n = 5$) and BVES-KO ($n = 6$) male mice. **e** The p62 and Cav3 immunostaining of GA muscles from 4-month-old WT and BVES-KO male mice. ($n = 5$ biologically independent experiments). Scale bar: 100 μm. **f, g,** Western blot (**f**) and quantification (**g**) of VPS34 in total GA muscle extracts from 8-month-old WT ($n = 5$) and BVES-KO ($n = 5$) male mice. **h, i** Live cell imaging (**h**) of RFP-LC3B and quantification (**i**) of LC3B puncta in FDB muscle fibers from 8-month-old WT ($n = 22$ fibers from 3 mice) and BVES-KO ($n = 22$ fibers from 3 mice) male mice at 5 days after electroporation with pCMV-RFP-LC3B. **j, k** Western blot (**j**) and quantification

(**k**) of LC3B/A in GA muscles from BVES-KO ($n = 4$ (without colchicine; $n = 3$ (with colchicine)) and WT ($n = 5$ each without/with colchicine) male mice without or with colchicine treatment as indicated (6 weeks of age). Blots are representative of two independent experiments. ns, not significant. One-way ANOVA with Turkey's multiple comparisons test. **l, m** Live cell imaging (**l**) of mCherry-GFP-LC3B dual reporter in FDB muscle fibers from 8-month-old WT ($n = 12$ fibers from 3 mice) and BVES-KO ($n = 12$ fibers from 3 mice) male mice at 5 days after electroporation and quantification (**m**) of red-only puncta normalized by total puncta. **n, o** Western blot (**n**) and quantification (**o**) of STX17 (S17) and VAMP7 (V7) normalized to GAPDH (G) in total GA muscle extracts from WT ($n = 5$) and BVES-KO ($n = 5$) male mice (8 months of age). Two-tailed unpaired Student's $t$ test was used for (**b–d, g, i, m, o**). Data are mean ± SEM. Source data are provided as a Source Data file.

flux may be due to downregulation of the SNARE proteins in BVES-KO skeletal muscles.

## Discussion

In summary, our work unveils a role of BVES in providing a negative feedback control for ADCY9 to regulate the cAMP signaling in skeletal muscle. The loss of BVES-mediated feedback control of the cAMP

signaling promotes PKA-mediated signaling cascade, dysregulates protein quality control systems and eventually leads to the loss of muscle integrity and function, highlighting the importance of this feedback control mechanism of cAMP in skeletal muscle. The different subcellular localization of BVES[4] implies that BVES may offer a potential feedback control mechanism for spatialized cAMP regulation. Since cAMP signaling participates in various physiological

processes in many different types of mammalian cells beyond skeletal muscle, defects in this feedback control mechanism may be linked to other human diseases.

In support of our finding that BVES inhibits ADCY9's activity, the BVES knockout mouse hippocampal neurons was found to display PKA-dependent enhancement of long-term potentiation[45]. The question remains as to how BVES regulates the activity of ADCY9. We speculate that BVES may inhibit ADCY9's activity in a cAMP concentration-dependent manner through the Popeye domain-mediated interaction. This is supported by a recent study showing that BVES interacts with ADCY9 via both the transmembrane domain and the C-terminal Popeye domain, although the authors did not study the regulation of ADCY9's activity by BVES[46]. Interestingly, the bimolecular fluorescence complementation (BiFC) assay showed that the Popeye domain interacted with ADCY9, but the co-IP failed to detect this interaction, suggesting that the interaction between the Popeye domain and ADCY9 is likely transient in nature. Another potential mechanism for BVES-mediated inhibition of ADCY9 is to regulate the membrane trafficking of ADCY9 in cAMP-dependent manner. In support of this possibility, BVES was shown to play a role in membrane protein trafficking[8]. Moreover, BVES was reported to interact with PDE4[47], suggesting that BVES may provide a delicate regulation of cAMP signaling through multiple interactions with the cAMP signaling machinery. Finally, it remains to be determined whether the structurally related BVES homologs POPDC2 and POPDC3 coordinate with BVES or independently regulate different ADCY proteins. Given the significance of genetic variations in *BVES*, *POPDC2* and *POPDC3* to human health, it will be important to examine how mutations in these genes contribute to muscular dystrophy, cardiac arrhythmia and other conditions, in regard to the dysregulated cAMP signaling, and determine whether manipulating the cAMP signaling could rescue the pathologies associated with their genetic defects.

Our data demonstrated that the loss of BVES-mediated cAMP feedback mechanism led to aberrant activation of UPS via a PKA-LKB1-AMPK-FoxO axis and treatment with the proteasome inhibitor bortezomib partially alleviated the muscle pathologies in BVES-KO mice, highlighting the therapeutic potential of targeting the cAMP/PKA-UPS signaling cascade for the treatment of BVES-related diseases. Our work also revealed that BVES-deficient skeletal muscle displayed a confound defect in autophagy execution. Previous studies showed that cAMP/PKA can either inhibit or activate autophagy depending on the cell or tissue context[28,48,49]. Exactly how BVES deficiency causes the autophagic defects and the relationship between the cAMP/PKA signaling and autophagy execution in skeletal muscle remain to be investigated. Our data suggested that downregulation of syntaxin 17 and VAMP7, which are involved in autophagosome fusion[4], potentially contributes to the observed defects in autophagy associated with BVES-KO skeletal muscles.

Consistent with our previous report[17], our data showed that fast-twitch fibers (IIb and IIx) were more severely affected in BVES-KO muscles. Interestingly, the AMPK phosphorylation was dramatically increased in the fast fiber-dominant EDL muscles from BVES-KO mice, but not in the slow fiber-containing soleus muscles (see Supplementary Fig. 8a). The exact mechanism for the fiber type-dependent effects in the BVES-KO mice remains to be determined. Future studies using transcriptomic (e.g. single nucleus RNA sequencing (snRNAseq)) and proteomic (e.g. immunoprecipitation-mass spectrometry) approaches with fast versus slow muscles may provide clues on the molecular and cellular mechanisms underlying the fiber type-dependent impacts caused by BVES deficiency.

## Methods

### Mice
All animal studies were reviewed and approved by the Institutional Animal Care and Use Committee (IACUC) of the Ohio State University.

C57BL/6N mice were purchased from The Jackson Laboratory (Bar Harbor, ME). The BVES-KO mice (C57BL/6N-Bvestm1.1(KOMP)Vlcg/MbpMmucd) with all coding exons deleted were obtained from Mutant Mouse Resource & Research Centers, UC Davis and maintained in our barrier facility. All mice were maintained under standard conditions of constant temperature (72 ± 4 °F), humidity (relative, 30–70%), in a specific pathogen-free facility and exposed to a 12-h light/dark cycle. The BVES-KO mice were genotyped by PCR analysis of genomic DNA prepared from ear clips with the following primers. The KO allele was amplified with a forward primer 5′-ACTTGCTT-TAAAAAACCTCCCACA and a reverse primer 5′-AGTCACTAGCAAGA-GATCTGCACCC and the WT allele was amplified using a forward primer 5′-AAGTGCTGGGATTAAAGGTGTGTGC and a reverse primer 5′-AAGGACACATCACAGCTTCAGG. The WT and KO allele would produce a 164-bp and 771-bp band, respectively.

### Plasmids
The pX601-stuffer-MHCK7-BVES-3xHA plasmid was constructed by subcloning the BVES fragment amplified from BVES-myc[50] and stuffer sequence from pLenti-hANO5WTint6BioID2[51] into XhoI and NotI digested pX601-AAV-CMV::NLS-SaCas9-NLS-3xHA-bGHpA;U6::BsaI-sgRNA, a gift from Feng Zhang (Addgene #61591). The pLVX-BVES-3xHA-puro was constructed by subcloning the BVES-3xHA into pLVX-puro (Clontech, San Jose, CA). The pCMV3-ADCY9-FLAG plasmid was purchased from Sino Biological (#HG19950-CF, Wayne, PA). pLVX-cAMPr-puro was constructed by subcloning the cAMPr fragment amplified from p2lox-cAMPr, a gift from Justin Blau (Addgene #99143)[29] into pLVX-puro. pCMV-AncBE4max was a gift from David Liu (Addgene #112094)[52]. The annealed gRNA oligos (targeting human *ADCY3* and *ADCY6*) were cloned into pLenti-OgRNA-Zeo plasmid as previously described[50]. pDEST-CMV-mCherry-GFP-LC3B WT was a gift from Robin Ketteler (Addgene #123230)[44]. pmRFP-LC3 was a gift from Tamotsu Yoshimori (Addgene #21075)[53].

### Cell culture and transfection
Cos1 and HEK293 cell lines were obtained from the American Type Culture Collection (ATCC). Cos1 and HEK293 cells were cultured in DMEM with 10% FBS. Transfection of HEK293 and COS-1 cells were performed using X-tremeGENE™ HP DNA transfection reagent (#6366244001, Sigma-Aldrich, St. Louis, MO). The cAMPr lentivirus was packaged in HEK293T cells by co-transfection with pLVX-cAMPr-puro, ΔNRF and pCMV-VSV-G according to our previous study[54]. The stable cAMPr-expressing HEK293 cells were established by lentiviral transduction, followed by puromycin selection.

### Generation of ADCY3/ADCY6 double KO HEK293 cells
To generate ADCY3/ADCY6 double KO HEK293 with stable expression of cAMPr, the stable cAMPr-expressing HEK293 cells were sorted for GFP into single cells in a 96-well plate at 24 h after transfection with the pCMV-AncBE4max, pLenti-ADCY3gRNA-Zeo and pLenti-ADCY6gRNA-Zeo. Sanger sequencing data were analyzed by using BEAT v1.0, which is published and available at https://github.com/Hanlab-OSU/Beat. The gRNA sequences are listed in the Supplementary Data 2. The expanded individual cell clones were screened by PCR and Sanger sequencing.

### Histology analysis and immunofluorescence staining
The skeletal muscles were embedded in optimal cutting temperature (OCT) compound, flash frozen using isopentane chilled in liquid nitrogen and kept at −80 °C until used. Cryosections were prepared using a cryostat Leica CM3054. For hematoxylin and eosin (H&E), transversely oriented sections (10 µm) were cut at mid-point, stained and imaged using a Nikon Ti-E inverted fluorescence microscope equipped with a Lumenera Infinity Color CCD camera and a Nikon Super Fluor 20×0.75 NA objective lens as well as INFINITY CAPTURE software v 6.5.0 (Nikon Inc., Melville, NY) as previously described[54].

For immunofluorescence staining, frozen tissue sections (10 μm) were fixed with 4% paraformaldehyde for 10 min at room temperature. After washing with PBS, the slides were blocked with 5% BSA/PBS for 1 h. The slides were incubated with the indicated primary antibodies (Supplementary Data 3) at 4 °C overnight. After extensive washing with PBS, the slides were incubated with the secondary antibodies (1:400) (Supplementary Data 3) for 1 h at room temperature. The slides were sealed with VECTASHIELD Antifade Mounting Medium with DAPI (Vector Laboratory, Burlingame, CA). The images were taken under a LSM780 microscope equipped with ZEN 2011 software (Zeiss, Germany) and assembled into figures using Adobe Illustrator 27.3.1 (Adobe, San Jose, CA). Fiber size and CNF quantification were carried out using Myosight[55] with manual calibration on the entire cross-sections of GA muscles (available at https://github.com/LyleBabcock/MyoSight/).

## In vivo protein synthesis measurements
In vivo protein synthesis was measured by using the SUnSET technique[39]. Briefly, WT and BVES-KO mice with 3-month-old were anesthetized and then given an intraperitoneal (I.P.) injection of 0.040 μmol/g puromycin (P8833, Sigma-Aldrich, St. Louis, MO) dissolved in 100 μl of PBS. At 30 min after I.P. injection, muscles were collected and frozen in liquid nitrogen for Western blot analysis. A mouse IgG2a monoclonal anti-puromycin antibody (clone 12D10, 1:5000) was used to detect puromycin incorporation.

## Intramuscular administration of AAV9-BVES into mice
AAV9 vectors were produced at Andelyn Biosciences (Columbus, OH) and titered by digital droplet PCR. AAV9-BVES viral particles (2 × 10$^{11}$ vg, 25 μl) were injected into the right gastrocnemius compartment of male BVES-KO mice at 4 months of age. At 1 and 3 months after AAV9 injection, force measurement (see below) was performed. Mice were sacrificed at 1 or 3 months after treatment. Gastrocnemius muscles were dissected out for morphometric analysis, immunofluorescence experiments and Western blot.

## Bortezomib treatment
Bortezomib was purchased from Selleck Chemicals (#S0130, Houston, TX). A stock solution (5 mg/ml) dissolved in the vehicle 2% DMSO, 30% PEG300 and ddH$_2$O were aliquoted and stored at 80 °C. An equal volume of Bortezomib (0.8 mg/kg diluted with the vehicle) or vehicle was I.P. injected once bi-weekly into BVES-KO mice ($N = 5$ for vehicle, $N = 6$ for bortezomib) at 12 weeks age, respectively. After 2 months of treatment, force measurement and exercise evaluation were performed. Mice were sacrificed after 3-month treatment. QU and GA muscles were processed for histopathology, immunofluorescence and Western blot analyses.

## Serum CK measurement
Blood samples collected from mice were transferred to MiniCollect tubes (Greiner Bio-one) and allowed to clot at room temperature for 30 min. Serum was prepared by centrifugation at 2350 × $g$ for 10 min and stored in −80 °C freezer for further use. Serum CK levels were measured using CREATINE KINASE-SL kit (326-10, Sekisui Diagnostics, Burlington, MA).

## Autophagic flux assay
A stock solution of colchicine (4 mg/ml in sterile ddH$_2$O, C9754, Sigma-Aldrich) was prepared and stored at −20 °C until the day of treatment. Immediately prior to administration, the colchicine stock solution was diluted to 0.1 mg/mL in sterile ddH$_2$O water. An equal volume of colchicine (0.4 mg/kg) or vehicle was administered to mice (6 weeks old) daily via I.P. injection for 7 days. Muscle tissues were harvested from treated mice on the day after the final treatment.

## Force measurement
The muscle contractility was measured using an in vivo muscle test system (Aurora Scientific Inc) as previously described[21]. Briefly, mice were anesthetized with 2% (w/v) isoflurane and anesthesia was maintained by 2% isoflurane (w/v) during muscle contractility measurement. Maximum plantarflexion tetanic torque was measured during a train of supramaximal electric stimulations of the tibial nerve (pulse frequency 150 Hz, pulse duration 0.2 ms) using the DMA v5.501 (Aurora Scientific Inc).

## Exercise and exhaustion assay
Time and distance to exhaustion was performed as previously described[56]. Briefly, prior to training, the randomized mice were firstly acclimated to the treadmill (LE8710MTS, Harvard Apparatus) for two consecutive days at slow speed (10 cm/s) for 5 min. On the third day, mice were placed on an uphill (15°) treadmill with an initial speed of 10 cm/s, increased every 4 min by 5 cm/s. Mice were considered to be exhausted when the animal's hindlimbs remained on the electric grid for more than 10 s. Time and distance were automatically collected via the software SeDaCOM v2.0 (Harvard Apparatus).

## Drop-out assay
The drop-out assay was performed according to the previous study[57]. Briefly, BVES-KO and age/sex-matched WT mice were initially trained (5 m/min running for 5 min each time, running for three times each day for three days) on treadmill. Then the mice were subjected to treadmill running at 10 m/min for 6 h. Twenty hours after the initial exercise training, mice were subjected to running at 6, 8, 10, 12, 14, and 16 m/min each for 3 min on the treadmill to test the capacity of recovery from muscle injury. The number of times the mice fail to run forward and touch the bottom of the electric grid of the treadmill and remain there for over 10 s was recorded as drop-out. Drop-outs of each mouse at each different speed were recorded.

## Voluntary running wheel
In this study, BVES-KO and age/sex-matched WT mice were individually housed in cages equipped with voluntary free-spinning running wheels (0297-0521, Columbus Instruments, Columbus, OH) for 9 days. The voluntary running activity were recorded by wheel rotations at 2 h intervals.

## Western blot
The cells and tissues were homogenized/lysed with cold RIPA buffer supplemented with protease inhibitors, and the extracted proteins were quantified by DC$^{TM}$ Protein Assay Reagent (Bio-Rad, Hercules, CA). The membrane extraction was performed using the Membrane Protein Extraction Kit (ab65400, Abcam, Waltham, MA) according to the manufacturer's instruction. The extracted protein samples were separated by stain-free SDS-PAGE gels (Bio-Rad, 4–15%) and transferred onto Nitrocellulose Membranes (0.45 μm). The membrane was incubated with the primary antibodies (Supplementary Data 3) at 4 °C overnight. Secondary HRP-conjugated goat anti-mouse (1:4000) and goat anti-rabbit (1:4000) antibodies were obtained from Cell Signaling Technology (Danvers, MA). The membranes were developed using ECL Western blotting substrate (Pierce Biotechnology, Rockford, IL) and images were taken on ChemiDoc XRS + system (BioRad, Hercules, CA). Western blots were quantified using Image Lab 5.2.1 software (Bio-Rad Laboratories, Hercules, CA) according to the manufacturer's instruction.

## RNA extraction and quantitative RT-PCR analysis
Total RNA was extracted from mouse tissues with Trizol. First-strand cDNA was synthesized using RevertAid RT Reverse Transcription Kit (K1691, Thermo Fisher Scientific, Waltham, MA). Real-time PCR was performed using PerfeCTa SYBR Green FastMix (95074, QuantaBio,

Beverly, MA) in QuantStudio™ 5 Real-Time PCR Systems with Quant-Studio Design & Analysis software v1.5.1 (Thermo Fisher Scientific). Samples were normalized for expression of *Gapdh* and analyzed by the $2^{-\Delta\Delta Ct}$ method.

### ChIP assay

The ChIP assay was performed as previously described[33], in BVES-KO skeletal muscles using the Magna ChIP A/G Chromatin Immunoprecipitation Kit (17-10085, Sigma-Aldrich). Soluble chromatin was co-immunoprecipitated with rabbit polyclonal anti-FoxO3 (1:200, #2497, Cell Signaling Technology) or an equal amount of control rabbit IgG (#2729, Cell Signaling Technology). After decrosslinking of the DNA, samples were subjected to quantitative PCR. The oligonucleotide primers used are shown in Supplementary Data 2.

### Immunoprecipitation and mass spectrum

The co-IP assay was performed as previously described[50] from Cos-1 cells co-transfected with ADCY9-FLAG and BVES-3xHA, using anti-Flag antibody (Sigma-Aldrich #F3165, 1:1000) and protein A/G agarose beads (#20423, Thermo Fisher Scientific). Gastrocnemius muscles from WT mice and BVES-KO mice with I.M. injection of AAV9-BVES-3xHA were lysed in the lysis buffer [25 mM Tris-HCl (pH 7.4), 150 mM NaCl, 5% Glycerol, 1% Triton X-100, 2 mM EDTA, and 1 mM DTT supplemented with protease inhibitor cocktail (Roche)]. Immunoprecipitation was performed by incubation with Pierce™ Anti-HA Magnetic Beads (#88836, Thermo Fisher Scientific) for 1 h at 4 °C. The beads were washed for four times with the lysis buffer. The immunoprecipitated samples were examined by Western blot and then subjected for mass spectrometric analysis at the Ohio State University Comprehensive Cancer Center Proteomic Shared Resources. Mass spectrum Data were searched against mouse database on MASCOT Via PD (ProteomeDiscoverer), and then analyzed using Scaffold 5.2.2.

### FDB muscle electroporation and isolation

The plasmids RFP-LC3B or mCherry-GFP-LC3B were transfected into FDB muscles by electroporation as described previously[58]. Briefly, the FDB muscle was injected with 10 μL of 2 mg/ml hyaluronidase solution (H4272, Sigma-Aldrich, St.Louis, MO). Two hours later, we injected a total of 20–50 μg of the indicated plasmids into the FDB muscles. After 15 min, the FDB muscles were electroporated using ECM630 (BTX, Holliston, MA) with the parameters (10 pulses, 20 ms in duration/each, 200 V). FDB muscle fibers were enzymatically isolated[59] at 6 days after electroporation and placed in Tyrode's solution (119 mM NaCl, 5 mM KCl, 25 mM HEPES, 2 mM CaCl2, 2 mM MgCl2, 6 g/L glucose, pH 7.4) for live cell imaging, which was taken using Zeiss 780.

### cAMPr reporter assay

The ADCY3/ADCY6 double KO HEK293 cells with stable expression of cAMPr were seeded in a 35-mm glass-bottom dish coated with collagen (A1048301, Thermo Fisher Scientific, Waltham, MA). The cells were transfected with the indicated vectors including pmCherry-C1, pCMV-ADCY9-Flag and pCMV-BVES-HA. At 24 h after transfection, the cells were washed twice with 1x Hanks' balanced salt solution and imaged under a Nikon Ti-E fluorescence microscope (Nikon, Melville, NY), equipped with a Zyla 5.5 sCMOS camera (Andor, Concord, MA) as well as NIS-Elements AR version 4.50 (Nikon, Melville, NY). Time series images were acquired every 4 s for 8 min. 500 μM IBMX was added to the culture dish at about 30 s after the fluorescence signal reaching a stable baseline. The cAMPr fluorescence signal data after background subtraction were plotted using Graphpad Prism 8.0.1.

### cAMP ELISA assay

The cAMP levels in skeletal muscles were measured using the cAMP Assay Kit (ab65355; Abcam, Waltham, MA) according to the manufacturer's instruction. Briefly, the 8-month-old WT and BVES-KO GA muscles were homogenized in 0.1 M HCl.

The amount of cAMP-HRP bound to the plate was determined by reading the colorimetric HRP activity at OD450 nm following sample acetylation. The measured cAMP values were normalized to the total protein content.

### Statistical analysis

Adobe Illustrator 27.3.1 was used to assemble figures. Sample size was estimated using G-power software 3.1. The data are expressed as mean ± the standard error of the mean (S.E.M.). Statistical differences were determined by two-tailed paired or unpaired Student's *t* test for two groups and one-way or two-way ANOVA with Turkey's post tests for multiple group comparisons using Prism 8.0.1 (Graphpad Software, La Jolla, California). A *P* value < 0.05 was considered to be significant.

### Reporting summary

Further information on research design is available in the Nature Portfolio Reporting Summary linked to this article.

## Data availability

The mass spectrometry data have been deposited in PRoteomics IDEntifications Database (https://www.ebi.ac.uk/pride/) under the accession number "PXD036346". All data generated or analyzed during this study are included within the article and its Supplementary Information files. Source data are provided with this paper.

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

## Acknowledgements

The authors thank the Analytical Cytometry Shared Resource of the Ohio State University Comprehensive Cancer Center for FACS and the Genomics Shared Resource of the Ohio State University Comprehensive Cancer Center for sequencing. R.H. is supported by US National Institutes of Health grant (R01HL116546) and a Parent Project Muscular Dystrophy award.

## Author contributions

R.H. conceived the study and wrote the manuscript. H.L. carried out the experiments, analyzed the data and drafted the manuscript. P.W., C.Z. Y.B.Z., and Y.Z. contributed to the plasmid preparation, reagent administration and mouse colony maintenance. All authors contributed to the final version of the manuscript.

## Competing interests

The Ohio State University has filed a provisional patent application (application number: 63/421,383; inventors: R.H. and H.L.) based on the gene therapy work and Bortezomib treatment work reported in this paper. The remaining authors declare no competing interests.
