## [Peer Review File · Nature Communications]

Defective BVES-mediated feedback control of cAMP in muscular dystrophyREVIEWER COMMENTS

Reviewer #1 (Remarks to the Author):

The manuscript by Li et al. aims to understand the role of BVES in control of cAMP signaling and muscle mass maintenance. BVES is known as a downstream effector of cAMP signaling. BVES mutations associate with muscular dystrophy and cardiac arrhythmia. However, the molecular functions of BVES remain incompletely understood. Using a global BVES knockout model, the authors examined the impact of BVES deficiency on muscle function, cAMP signaling, UPS function, protein synthesis, and autophagy function. Based on the collected data, the authors propose that BVES is a negative regulator of cAMP signaling by interacting with ADCY9, and that loss of BVES promotes cAMP-PKA signaling, enhances UPS function, impairs autophagic flux, and consequently, muscle atrophy and weakness.

Overall, this is an elegant study with logical and rigorous experiment designs and careful experimental execution. The data are of high quality, convincing and in general, support the conclusions. The findings are novel and provide new insights into the biological and molecular functions of BVES in skeletal muscle.

This reviewer identifies one major question related to the role of BVES in regulating UPS function. Addressing it may improve the quality of the manuscript.

1. The accumulation of ubiquitinated proteins in skeletal muscle (Fig. 5f) could be explained by enhanced ubiquitination and/or impaired proteasome function. Given the reported role of cAMP-PKA signaling in proteasome function, does loss of BVES affect the proteasomal activities before and after muscular dysfunction?
2. Fig. 5f, the authors can probe the abundance of K48- and K63-linked ubiquitinated proteins to strengthen the effect of BVES-KO on protein ubiquitination.
3. The evidence supporting the activation of Foxo proteins by BVES-KO is a bit weak. ChIP assay and/or luciferase assay could be performed to assess Foxo activities. Further, does the blockade of cAMP-PKA signaling affect the expression/activity of Foxo and its downstream targets (MuRF, Atrogin 1)?

Reviewer #2 (Remarks to the Author):

Review

The present study aims to determine the feedback mechanisms for cAMP signaling control in skeletal muscles. The blood vessel epicardial substance (BVES) is a negative regulator of ADCY9-mediated cAMP signaling in maintaining skeletal muscle mass and function. The authors used BVES knockout mice to demonstrate the role of BVES in skeletal muscle mass and function. BVES negatively regulates adenylyl

cyclases, promoting the FoxO-mediated ubiquitin-proteasome degradation pathway and autophagy initiation. Overall, it is a novel and innovative study to define the role of BVES in skeletal muscles.

Major concerns

- BVES-KO caused type2b fiber-specific atrophy (Fig2-l) and damage, but the authors did not address why/how the KO impacted other types of muscle/fiber. They need to present BVES protein expression in slow and fast twitch muscles.
- In Figures 6h-i and l-m, they tested autophagic activities by measuring LC3B puncta on FDB fibers, but type2b fibers are rarely found in FDB. The fiber they used could be either type2a or 2x, which was not atrophied by BVES KO. They must address how LC3B signals increase these fiber types (type2a or thep2x).

Minor comments

- Missing details about histologic analyses, number of mice, muscles, slides, and areas of interest they observed.
- As the authors presented, BVES is expressed in multiple organs, including heart and smooth muscle. So whole body complications (e.g. heart mass reduced in KO) are also attributable to the functional loss and atrophy of skeletal muscle. Similarly, AAV induced BVES overexpression in the KO muscle may improve the function of vessel and nerve. This is also supported by the results shown in Fig 3d-j. After AAV-injection (BVES), muscle mass and force are almost fully recovered, but fiber CSA is still low, and percent CLFs is over 10%. This uncoupling of mass and force from muscle morphology and structure indicates that other tissues/cells contribute to the mass and function recovery.
- Male and female balance in WB and histology need to be addressed, as muscle and heart weight changes differ between males and females.
- In Fig. 5h and i, FOXO3 translocation images are not convincing as higher resolution and quantification are required. Also, I am curious why FOXO3 is highly expressed in central nuclei, indicative of regenerating fiber.

Reviewer #3 (Remarks to the Author):

What are the noteworthy results?

The most relevant results are: identification of the relationship between muscle impairment and BVES disruption, demonstration of the diverse impact of BVES disruption in different muscles, identification of the relationship between BVES and cAMP signaling and induction of atrophy by PKA activation.

Will the work be of significance to the field and related fields?

It is significant for muscle research since it demonstrate the close relationship between BVES disruption and atrophy analyzing step by step all markers known to be involved in muscle atrophy but not directly associated previously with BVES.

How does it compare to the established literature? If the work is not original, please provide relevant references.

Revising the recent literature, the disruption of BVES in muscle disorders has been described but not molecularly characterized. The authors carefully demonstrate, step by step all the major events involved in this disruption. The treatment with bortezomib supported the mechanistic understanding of molecules involved in this atrophic process and offers the possibility to be utilized in patients .

Does the work support the conclusions and claims, or is additional evidence needed?

The work supports the conclusions and claim, but what is missing is the explanation of the different impact of BVES on soleus muscle, it will be great if the authors can get a better insight into this issue: why the soleus behaves differently? Could be related with different expression of key molecules involved in the pathogenesis differently expressed in this muscle? I believe that few experiments very easy to perform will provide a more detailed definition of what is occurring in different muscles.

Are there any flaws in the data analysis, interpretation and conclusions?

No.

Do these prohibit publication or require revision?

No

Is the methodology sound?

Yes

Does the work meet the expected standards in your field?

Yes

Is there enough detail provided in the methods for the work to be reproduced?

Yes

Reviewer #1 (Remarks to the Author):

The manuscript by Li et al. aims to understand the role of BVES in control of cAMP signaling and muscle mass maintenance. BVES is known as a downstream effector of cAMP signaling. BVES mutations associate with muscular dystrophy and cardiac arrhythmia. However, the molecular functions of BVES remain incompletely understood. Using a global BVES knockout model, the authors examined the impact of BVES deficiency on muscle function, cAMP signaling, UPS function, protein synthesis, and autophagy function. Based on the collected data, the authors propose that BVES is a negative regulator of cAMP signaling by interacting with ADCY9, and that loss of BVES promotes cAMP-PKA signaling, enhances UPS function, impairs autophagic flux, and consequently, muscle atrophy and weakness.

Overall, this is an elegant study with logical and rigorous experiment designs and careful experimental execution. The data are of high quality, convincing and in general, support the conclusions. The findings are novel and provide new insights into the biological and molecular functions of BVES in skeletal muscle.

Response: We thank the reviewer 1 for sharing the excitement of our work.

This reviewer identifies one major question related to the role of BVES in regulating UPS function. Addressing it may improve the quality of the manuscript.

1. The accumulation of ubiquitinated proteins in skeletal muscle (Fig. 5f) could be explained by enhanced ubiquitination and/or impaired proteasome function. Given the reported role of cAMP-PKA signaling in proteasome function, does loss of BVES affect the proteasomal activities before and after muscular dysfunction?

A: To determine whether BVES loss affects the proteasomal activities before and after disease onset, we examined the ubiquitination levels in skeletal muscles from 2- and 8-month-old WT and BVES-KO mice because we observed that muscular dystrophy and dysfunction started at ~3 months of age in BVES-KO mice. As shown in **Supplementary Figure 10a, b** (2-month-old) and **Figure 5f, g** (8-month-old WT), ubiquitination was significantly increased in BVES-KO muscles at both ages. Combined with the rescue effect of the proteasome inhibitor bortezomib on muscular dystrophy caused by BVES deficiency, we concluded that the loss of BVES increased ubiquitination for proteasomal degradation before and after muscular dysfunction.

2. Fig. 5f, the authors can probe the abundance of K48- and K63-linked ubiquitinated proteins to strengthen the effect of BVES-KO on protein ubiquitination.

A: Thank you for the suggestion. Ubiquitin contains seven lysines (K6, K11, K27, K29, K33, K48 and K63) and ubiquitin molecules can be linked through any of these seven lysines. We examined the abundance of K48- and K63-linked ubiquitinated proteins in 8-month-old WT and BVES-KO skeletal muscles as suggested. We did not observe major differences in either K48 or K63-linked ubiquitination (**Supplementary Figure 10c**). Thus, BVES deficiency may instead affect other lysines such as K11 and K29-linked ubiquitination.

3. The evidence supporting the activation of Foxo proteins by BVES-KO is a bit weak. ChIP assay and/or luciferase assay could be performed to assess Foxo activities. Further, does the blockade of cAMP-PKA signaling affect the expression/activity of Foxo and its downstream targets (MuRF, Atrogin 1)?

Response: Thank you for this insightful question. First, we injected the specific PKA inhibitor H89 (2 mg/kg) into skeletal muscle, which effectively suppressed the PKA activity (**Supplementary Figure 11a**). Next, we performed the CHIP assay to determine if H89 can affect the binding of FoxO to the regulatory elements of its downstream targets. As shown in **Supplementary Figure 11b, c**, H89 significantly suppressed the binding of FoxO3a to MuRF1 and Atrogin-1 promoters in BVES-KO muscles, further supporting our conclusion that FoxO signaling activation is involved in the pathogenesis of BVES-deficient muscular dystrophy.

Reviewer #2 (Remarks to the Author):

Review

The present study aims to determine the feedback mechanisms for cAMP signaling control in skeletal muscles. The blood vessel epicardial substance (BVES) is a negative regulator of ADCY9-mediated cAMP signaling in maintaining skeletal muscle mass and function. The authors used BVES knockout mice to demonstrate the role of BVES in skeletal muscle mass and function. BVES negatively regulates adenylyl cyclases, promoting the FoxO-mediated ubiquitin-proteasome degradation pathway and autophagy initiation. Overall, it is a novel and innovative study to define the role of BVES in skeletal muscles.

Response: Thank you very much for stating the novelty of our study.

Major concerns

- BVES-KO caused type2b fiber-specific atrophy (Fig2-l) and damage, but the authors did not address why/how the KO impacted other types of muscle/fiber. They need to present BVES protein expression in slow and fast twitch muscles.

Response: A previous study showed that soleus is composed of 75% slow-twitch fibers whereas EDL has more than 90% fast-twitch fibers (J Physiol. 2004 Sep 1; 559(Pt 2): 519–533). Thus, we examined the expression of BVES in soleus and EDL muscles. As showed in **Supplementary Figure 8c, d**, the expression of BVES was not significantly different between soleus and EDL in WT mice. We have performed additional experiments to shed light into why the KO impacted preferentially fast-twitch muscle fibers. Western blot showed that AMPK was significantly activated in EDL from KO mice compared with that from WT mice, while AMPK phosphorylation was not significantly affected in soleus muscle of KO mice (**Supplementary Figure 8a, b**), suggesting that other regulatory mechanisms may exist in slow fibers to compensate for BVES deficiency, which will be investigated in our future study.

- In Figures 6h-i and l-m, they tested autophagic activities by measuring LC3B puncta on FDB fibers, but type2b fibers are rarely found in FDB. The fiber they used could be either type2a or 2x, which was not atrophied by BVES KO. They must address how LC3B signals increase these fiber types (type2a or thep2x).

Response: Although type 2b fibers are rarely found in FDB muscles, previous work showed that FDB muscle contains ~50% type 2x muscle fibers (<https://skeletalmusclejournal.biomedcentral.com/articles/10.1186/s13395-018-0160-3>). We

performed the immunostaining using anti-MyHC-2a and anti-MyHC-1 as well as anti-dystrophin and found that the type 2x fibers (negative for MyHC-2a and MyHC-1) were also significantly smaller in BVES-KO mice compared to those in WT mice (**Supplementary Figure 5**), suggesting that type 2x muscle fibers are also affected in BVES-KO mice. It is of note that myosin heavy chain isoform 2x rather than 2b is expressed in human skeletal muscle, and that both type 2x and 2b are very similar in their contractile properties and oxidative capacity (Mori, *Nutrients*, 13(5):1538.).

Minor comments

- Missing details about histologic analyses, number of mice, muscles, slides, and areas of interest they observed.

Response: Thank you. We have now added the details about the number of mice, muscles, slides, and areas of interest for our experiments.

- As the authors presented, BVES is expressed in multiple organs, including heart and smooth muscle. So whole body complications (e.g. heart mass reduced in KO) are also attributable to the functional loss and atrophy of skeletal muscle. Similarly, AAV induced BVES overexpression in the KO muscle may improve the function of vessel and nerve. This is also supported by the results shown in Fig 3d-j. After AAV-injection (BVES), muscle mass and force are almost fully recovered, but fiber CSA is still low, and percent CLFs is over 10%. This uncoupling of mass and force from muscle morphology and structure indicates that other tissues/cells contribute to the mass and function recovery.

Response: First of all, the AAV is injected intramuscularly and the promoter used is a muscle/heart specific promoter (MHCK7), which together restricts the expression of the BVES transgene into adult muscle fibers of the injected muscle.

Moreover, the force and mass shown in Fig. 3d-j were normalized to the body mass, which was still low as compared to wild type because the AAV.BVES was only administered into one single GA muscle. Therefore, it's not proper to compare these relative parameters in AAV-treated muscle to the ones in WT mice (and claim as "fully recovered"). Indeed, if we plot the absolute values of muscle mass from the AAV-treated GA muscle, these values are still lower than the ones in the WT mice (**Supplementary Figure 6**).

- Male and female balance in WB and histology need to be addressed, as muscle and heart weight changes differ between males and females.

Response: All the WB and histology studies were performed separately in male and female, and the gender was clearly specified in the figure legends.

- In Fig. 5h and i, FOXO3 translocation images are not convincing as higher resolution and quantification are required. Also, I am curious why FOXO3 is highly expressed in central nuclei, indicative of regenerating fiber.

Response: We have provided the immunofluorescence images of FOXO3 in high resolution and

quantified the FOXO3+ nuclei per field (**Figure 5h-k**). Indeed, we observed that central nuclei were high in the FOXO3 signal, indicating the regenerating muscle fibers may have higher FoxO3 nuclear translocation. Although beyond the scope of this study, understanding the regulation of FoxO3 nuclear translocation in regenerating muscle fibers by BVES may help to develop strategies to improve muscle regeneration.

Reviewer #3 (Remarks to the Author):

What are the noteworthy results?

The most relevant results are: identification of the relationship between muscle impairment and BVES disruption, demonstration of the diverse impact of BVES disruption in different muscles, identification of the relationship between BVES and cAMP signaling and induction of atrophy by PKA activation.

Response: Thank you very much for the precise summary of the major results in our study.

Will the work be of significance to the field and related fields?

It is significant for muscle research since it demonstrates the close relationship between BVES disruption and atrophy analyzing step by step all markers known to be involved in muscle atrophy but not directly associated previously with BVES.

Response: Thank you very much for the endorsement of the significance of our study.

How does it compare to the established literature? If the work is not original, please provide relevant references.

Revising the recent literature, the disruption of BVES in muscle disorders has been described but not molecularly characterized. The authors carefully demonstrate, step by step all the major events involved in this disruption. The treatment with bortezomib supported the mechanistic understanding of molecules involved in this atrophic process and offers the possibility to be utilized in patients.

Response: Thank you very much for recognizing the novelty of our study.

Does the work support the conclusions and claims, or is additional evidence needed?

The work supports the conclusions and claim, but what is missing is the explanation of the different impact of BVES on soleus muscle, it will be great if the authors can get a better insight into this issue: why the soleus behaves differently? Could be related with different expression of key molecules involved in the pathogenesis differently expressed in this muscle? I believe that few experiments very easy to perform will provide a more detailed definition of what is occurring in different muscles.

Response: Please see our response above to Reviewer 2.

REVIEWERS' COMMENTS

Reviewer #1 (Remarks to the Author):

The authors addressed my concerns. I do not have further comments.

Reviewer #2 (Remarks to the Author):

The revised manuscript has been improved satisfactorily. The authors also have addressed all of my concerns. So I strongly recommend that the manuscript should be accepted for publication in the present form. Thank you very much for giving me the opportunity to review this article for Nature Communications.

Reviewer #3 (Remarks to the Author):

The paper presents the identification of mechanisms associated with BVES and muscle impairment in a KO mouse model identifying the relationship between BVES, cAMP signaling and induction of atrophy by PKA activation. It has been carefully revised by Authors and improved.

Specifically, the authors dissect step by step all markers known to be involved in muscle atrophy in BVES-KO model and bortezomib treatment and AAV9.BVES transfected animals are able to counteract atrophy and muscle impairment. In this context, by dissecting molecular mechanisms of protein degradation machinery (autophagy and lysosomal degradation) the Authors demonstrate that PKA signalling activation promotes FOXO mediated protein ubiquitination enhancing autophagy and lysosomal degradation.

Revising the recent literature, Limb-girdle muscular dystrophy carrying a mutation in BVES (LGMDR25) could take advantage from treatment with recombinant AAV9.BVES. The recovery of the muscle mass and function was described and published (Molecular Therapy, February 2023) by the same Authors. The treatment dramatically improved body weight gain, muscle mass, muscle strength, and exercise performance in BVES-KO mice (both in male and female mice) supporting results described in the present manuscript. They confirm that quadriceps and tibialis anterior muscles were more affected with a higher number of central nuclei fibers as compared with soleus and diaphragm in BVES-KO mice. This could suggest that a molecular characterization by immunoprecipitation and mass spectrometry analysis

adopting a similar protocol described in the present paper could provide hints to clarify this point. I believe a comment on this point at least as future perspective should be included.

The data support the conclusions and claim.

We are grateful to the reviewers for their timely review of our manuscript and glad to hear that all reviewers are happy with our responses to their previous critiques. Below is a point-by-point response to each individual comments.

Reviewer #1 (Remarks to the Author):

The authors addressed my concerns. I do not have further comments.

Response: We appreciate the Reviewer #1 for the time to review our revised manuscript. We are happy to hear that all of Reviewer #1's concerns have been addressed.

Reviewer #2 (Remarks to the Author):

The revised manuscript has been improved satisfactorily. The authors also have addressed all of my concerns. So I strongly recommend that the manuscript should be accepted for publication in the present form. Thank you very much for giving me the opportunity to review this article for Nature Communications..

Response: Thank you very much! We are happy to hear that all of Reviewer #2's concerns have been addressed.

Reviewer #3 (Remarks to the Author):

The paper presents the identification of mechanisms associated with BVES and muscle impairment in a KO mouse model identifying the relationship between BVES, cAMP signaling and induction of atrophy by PKA activation. It has been carefully revised by Authors and improved.

Specifically, the authors dissect step by step all markers known to be involved in muscle atrophy In BVES-KO model and bortezomib treatment and AVV9.BVES transfected animals are able to counteract atrophy and muscle impairment. In this context, by dissecting molecular mechanisms of protein degradation machinery (autophagy and lysosomal degradation) the Authors demonstrate that PKA signalling activation promotes FOXO mediated protein ubiquitination enhancing autophagy and lysosomal degradation.

Response: Thank you very much for the positive statement about our revised manuscript.

Revising the recent literature, Limb-girdle muscular dystrophy carrying a mutation in BVES (LGMDR25) could take advantage from treatment with recombinant AAV9.BVES. The recovery of the muscle mass and function was described and published (Molecular Therapy, February 2023) by the same Authors. The treatment dramatically improved body weight gain, muscle mass, muscle strength, and exercise performance in BVES-KO mice (both in male and female mice) supporting results described in the present

manuscript. They confirm that quadriceps and tibialis anterior muscles were more affected with a higher number central nuclei fibers as compared with soleus and diaphragm in BVES-KO mice. This could suggest that a molecular characterization by immunoprecipitation and mass spectrometry analysis adopting a similar protocol described in the present paper could provide hints to clarify this point. I believe a comment on this point at least as future perspective should be included. The data support the conclusions and claim.

Response: Thank you very much for the suggestion. We have added the following paragraph into the Discussion:

Consistent with our previous report⁵¹, our data showed that fast-twitch fibers (IIb and IIx) were more severely affected in BVES-KO muscles. Interestingly, the AMPK phosphorylation was dramatically increased in the fast fiber-dominant EDL muscles from BVES-KO mice, but not in the slow fiber-containing soleus muscles (see **Supplementary Fig. 8a**). The exact mechanism for the fiber type-dependent effects in the BVES-KO mice remains to be determined. Future studies using transcriptomic (e.g. single nucleus RNA sequencing (snRNAseq)) and proteomic (e.g. immunoprecipitation-mass spectrometry) approaches with fast versus slow muscles may provide clues on the molecular and cellular mechanisms underlying the fiber type-dependent impacts caused by BVES deficiency.